# Shaping the mid-Miocene warmth: a sensitivity study on paleogeography, CO<sub>2</sub> and model physics

Martin Renoult<sup>1,2</sup>, Agatha de Boer<sup>1,2</sup>, and Ellen Berntell<sup>2,3</sup>

Correspondence: Martin Renoult (martin.renoult@misu.su.se)

Abstract. The mid-Miocene (15.98 to 13.82 Ma) was a period characterized by substantially warmer temperatures than today and atmospheric CO2 concentrations comparable to near-future projections. Climate models have generally struggled to reproduce proxy-based reconstructions from this interval, particularly at high latitudes where model temperatures are consistently too low. Here, we present new mid-Miocene simulations using an unpublished geography and evaluate the climate's sensitivity to several key components: paleogeography (including land-sea distribution, topography and ice sheets), atmospheric CO<sub>2</sub> concentration, atmospheric model choice, and solar forcing. Our baseline mid-Miocene climate yields a global mean surface temperature (GMST) of 19.8°C. GMST varies by up to 3.2°C between simulations with CO<sub>2</sub> concentrations of two and four times pre-industrial values, which is consistent with estimates for the mid-Miocene. Removal of the Antarctic ice-sheet leads to expected local warming, but nevertheless records an overall global cooling of 1.3 C. Solar forcing and subtle changes of land-sea mask each impact GMST by around 0.2°C. The choice of atmospheric model substantially affects the simulated mid-Miocene climate through modified feedback mechanisms. We estimate an equilibrium climate sensitivity (ECS) of 2.9°C for the mid-Miocene, similar to modern-based estimates from our model, indicating the potential for the Miocene to contribute to constraining equilibrium ECS. Global precipitation is tightly coupled to GMST across all our simulation. As with previous studies, all our simulations, regardless of specific configuration, underestimate high-latitude proxy-reconstructed temperatures. This highlights the need to improve our understanding on polar amplification and the need to use high concentrations of CO<sub>2</sub> to compensate for a cold modeled Miocene climate.

#### 1 Introduction

The Langhian (15.98 to 13.82 Ma) is a critical stage of the Miocene marked by the warm event known as the Mid-Miocene Climatic Optimum (MMCO), when global mean surface temperatures (GMST) were around 7.6°C warmer than today (Zachos et al., 2001; Goldner et al., 2014; Burls et al., 2021). During this time, tropical gateways connected the Pacific, Atlantic

<sup>&</sup>lt;sup>1</sup>Department of Geological Sciences, Stockholm University, Stockholm, Sweden

<sup>&</sup>lt;sup>2</sup>Bolin Center for Climate Research, Stockholm, Sweden

<sup>&</sup>lt;sup>3</sup>Department of Meteorology, Stockholm University, Stockholm, Sweden

and Indian oceans, while polar gateways connecting the Arctic with the Atlantic were more restricted than today and the Arctic-Pacific connection was closed off. This epoch further saw a substantial change in flora and the emergence of modern biomes (Steinthorsdottir et al., 2021).

Warm paleoclimates are often investigated as potential analogs for near-future climates (Yin and Berger, 2015; Burke et al., 2018). While the analog nature of past climate are never perfect, and there are many differences in how warming manifests itself in past and future climate states (Sicard et al., 2023; Oldeman et al., 2024), the past climates offer a comparison of modelling outcomes with proxy data, which in turn can provide insights into climate processes that also act in the future. An example is constraining the important climate change metric of equilibrium ECS (ECS), the long-term global mean surface temperature change in response to a doubling of CO<sub>2</sub> from its pre-industrial level, as it was previously done with the Pliocene (Hargreaves and Annan, 2016; Haywood et al., 2020; Sherwood et al., 2020; Renoult et al., 2020, 2023).

In this context, the mid-Miocene has recently emerged as a particularly exciting testbed for understanding warm climates. In contrast to the more recent interglacials and the warm period of the mid-Pliocene, or the older Eocene period, mid-Miocene atmospheric CO<sub>2</sub> concentrations are thought to be close to modern values and near-future estimates (400 to 800 ppm), and potentially peaking at much higher values during the MMCO (Rae et al., 2021; Steinthorsdottir et al., 2025). As such, it has been the subject of several recent modelling and proxy syntheses studies (Steinthorsdottir et al., 2021; Burls et al., 2021; Naik et al., 2025), and a first formal Miocene Model Intercomparison Project (MioMIP) is under design.

Climate models have underestimated proxy-reconstructed temperatures of the mid-Miocene at equivalent CO<sub>2</sub> concentrations (Goldner et al., 2014; Burls et al., 2021), in particular at high latitudes (e.g. Herold et al., 2011). This is not a specific issue of the Miocene, as polar amplified warmth is also weak in other warm paleoclimates, such as the Pliocene (Haywood et al., 2020) and the Eocene (Lunt et al., 2021). This suggests a misrepresentation of high latitude climate feedbacks in models, substantial uncertainties on climate forcings, and/or a persistent bias in high latitude proxies. These points are critical to address, as they can allow to improve simulations of the Miocene, as well as model processes and the accuracy of future model predictions.

The aim of this study is to explore the mid-Miocene climate and investigate its sensitivity around various uncertain boundary conditions and model parametrizations. We perform simulations with a new, unpublished paleogeography of the early Mid-Miocene, specifically the Langhian, and compare the simulation to one in the same model using the published paleogeography of (Burls et al., 2021). We also perform simple computations of Miocene ECS as to evaluate its potential as a new line of evidence for constraining ECS, in the light of the upcoming MioMIP. We further explore the sensitivity to the Antarctic ice-sheet existence, CO<sub>2</sub> concentrations, the solar constant, and the atmospheric model.

#### 50 2 Methods

We use the Community Earth System Model version 1.2 (CESM1.2) to simulate Langhian or mid-Miocene climate states. CESM1.2 is a coupled model using the ocean model POP2, with a 1° horizontal grid and 60 vertical levels and the CAM4 model for the atmosphere, applying a horizontal grid of 1.9x2.5° with 26 vertical layers, as well as a sea-ice model (CICE4),

**Table 1.** Summary of the experiments carried out in this study. "Getech" refers to the Langhian geography provided by Getech Plc., whereas Burls et al. (2021) is a mid-Miocene geography. \*The simulations were ran for 150 years as to estimate ECS.

| Experiment       | Geography           | CO <sub>2</sub> (ppm) | Solar constant (Wm <sup>-2</sup> ) | Ice sheet           | Atmospheric model | Initialization |
|------------------|---------------------|-----------------------|------------------------------------|---------------------|-------------------|----------------|
| PI               | Modern              | 284.7                 | 1360.89                            | Modern              | CAM4              | Default        |
| PI_DblCO2*       | Modern              | 569.4                 | 1360.89                            | Modern              | CAM4              | PI             |
| Mio_Ctrl         | Getech              | 854.1                 | 1359.51                            | Getech              | CAM4              | Default        |
| Mio_2x           | Getech              | 569.4                 | 1359.51                            | Getech              | CAM4              | Default        |
| Mio_4x           | Getech              | 1138.8                | 1359.51                            | Getech              | CAM4              | Default        |
| Mio_Ctrl_DblCO2* | Getech              | 1708.2                | 1359.51                            | Getech              | CAM4              | Mio_Ctrl       |
| Mio_2x_DblCO2*   | Getech              | 1138.8                | 1359.51                            | Getech              | CAM4              | Mio_2x         |
| Mio_BurlsGeo     | Burls et al. (2021) | 854.1                 | 1360.89                            | Burls et al. (2021) | CAM4              | Default        |
| Mio_noIS         | Getech              | 854.1                 | 1359.51                            | None                | CAM4              | Default        |
| Mio_noIS_solPI   | Getech              | 854.1                 | 1360.89                            | None                | CAM4              | Default        |
| Mio_noIS_CAM5    | Getech              | 854.1                 | 1359.51                            | None                | CAM5              | Default        |

a river runoff model (RTM) and a land model which includes the carbon-nitrogen cycle and dynamical vegetation (CLM2). CESM1.2 has been used in numerous studies simulating different paleoclimates, such as the LGM (Kageyama et al., 2021), the Pliocene (Haywood et al., 2020), the Eocene (Lunt et al., 2021) as well as older paleoclimates (Li et al., 2022).

We perform a "best-to-knowledge" control simulation of the Langhian (Mio\_Ctrl), which includes an Antarctic ice sheet reconstruction, an updated CO<sub>2</sub> estimate, Langhian solar constant, and is based on a topography and bathymetry reconstruction of the Langhian, provided by Getech Plc. We compare this new paleogeography to an identical simulation using the paleogeography of MioMIP2 Phase 1 (Burls et al., 2021) which is provided in Fig. 1. The extent of Arctic and Antarctic ice sheets between Mio\_Ctrl and Mio\_BurlsGeo is compared to pre-industrial (PI) in Fig 2. The ice sheets are considered as topographic objects and therefore the topography is flattened in Antarctica in the experiments where the ice sheets are not included. We then compare several Miocene simulations that differ in their solar constant (Mio\_noIS\_solPI), the presence of Antarctic ice sheet (Mio\_noIS), and the atmospheric model used (Mio\_noIS\_CAM5), as to identify the role of each onto the Langhian climate state. A summary of each experiment is given in Table 1. A comparison of the Langhian paleogeography of Getech Plc. and the paleogeography of MioMIP2 Phase 1 (Burls et al., 2021)

Unless specified for our sensitivity tests, our simulations use an atmospheric CO<sub>2</sub> concentration of 854.1 ppm, which is equivalent to three times PI CO<sub>2</sub> concentration of the model. This is planned as the standard concentration for the protocol of the MioMIP-phase 2 project. It is within the range of some reconstructions provided for the mid-Miocene (Rae et al., 2021; Steinthorsdottir et al., 2021), and it has also been used in other mid-Miocene simulations, which in general provides a closer match to proxy temperatures than simulations with lower CO2 concentrations (Burls et al., 2021). The solar constant is calculated from a model of solar luminosity increase with time (Gough, 1981), which leads to a value of 1359.51 W.m<sup>-2</sup> for the Langhian, in comparison to 1360.89 W.m<sup>-2</sup> at PI. Aerosols are kept as PI, but the sources are moved to match with the

Figure 1. Topography and bathymetry for the two main paleogeographies in our simulations: A) paleogeography of Mio\_Ctrl, using Getech Plc. and B) paleogeography of Mio\_BurlsGeo based on Burls et al. (2021). C) is the difference in topography between both paleogeographies. Except Mio\_BurlsGeo, all simulations use A) Getech Plc.

Langhian paleogeography (for instance if an ocean turned to land). Non-CO2 greenhouse gases and orbital forcing are kept as pre-industrial. Our simulations are compared to a proxy record presented in Burls et al. (2021). It is an agglomeration of sea-surface and surface temperatures based on multi-proxy reconstructions, where the sites are then rotated onto the Langhian geography using the tectonic model of Getech Plc.

For estimating Miocene ECS in our simulation, we base our approach on a variation of Gregory et al. (2004), which would require an equilibrium simulation of the mid-Miocene with PI CO<sub>2</sub> concentration, from where a new simulation is initialized with an abrupt and sustained quadrupling of the CO<sub>2</sub> concentration, for around 150 years, so that "fast" climate feedbacks have mostly responded to the initial forcing (i.e. clouds, surface albedo, water vapor, lapse-rate and Planck feedbacks). In the case of our simulations, a mid-Miocene state with PI CO<sub>2</sub> is numerically unstable, therefore we initialize from Mio\_Ctrl by instead abruptly doubling its CO<sub>2</sub> concentration (Mio\_Ctrl\_DblCO<sub>2</sub>), as to avoid reaching excessively high CO<sub>2</sub> concentrations under a quadrupling, which could introduce non-linear effects on climate feedbacks. The ECS is then extrapolated from the global mean annual surface temperature after 150 years of simulation minus the initial temperature of the simulation using ordinary least squares linear regression. As to roughly estimate the influence of the Miocene background temperature on ECS, we also

Figure 2. Land ice extent (filled contours) in the Arctic (top row) and the Antarctic (bottom row) regions in the three geographies used in this study: A) PI, B) Mio\_Ctrl with the paleogeography of Getech and C), Mio\_BurlsGeo with the paleogeography of Burls et al. (2021). Note that the PI geography also includes high altitude glaciers, which are not all shown. Land ice shown as extending over the ocean is only an artifact of interpolation, as land ice is limited to land areas.

apply the same method but from a simulation at lower  $CO_2$  concentration,  $Mio_2x$  ( $Mio_2x_DblCO_2$ ), and we compare it to a PI simulation with an abrupt doubling of  $CO_2$  (PI\_DblCO\_2).

The model is initialized from a warm ocean state, where the average deep-ocean temperature is around 6.6°C. This value comes from an earlier, unpublished mid-Miocene simulation performed with CESM1.2, and is used to accelerate our deep-ocean equilibrium. The salinity is initialized globally at 35 PSU. In South America, in the so-called Lake Pebas, salinity becomes negative due to excessive precipitation and river runoff in these warm conditions. Therefore, we apply a marginal sea balancing scheme which redistributes the gain of freshwater onto the world oceans to avoid the spread of nonphysical, negative salinity into the oceans. This scheme has previously been used with CESM1.2 in warm paleoclimates (e.g. Lunt et al., 2021). The sea-ice model is initialized ice-free and the dynamic vegetation is initialized as bare soil. All runs have been integrated for at least 3100 years and up to 5000 years when using CAM5 (experiment Mio\_noIS\_CAM5), starting from default initial conditions, and the last 100 years are averaged and used for the results.

Several aspects of the ocean model are modified to handle the large changes in land-sea mask and bathymetry at the Langhian. Overflows and tidal mixing are turned off, as they are hard-coded on a pre-industrial ocean grid. We follow the recommendations for deep-time paleoclimates of NCAR to increase ocean background vertical diffusivity, to compensate for the lack of tidal mixing (found at https://www2.cesm.ucar.edu/models/paleo/faq/, last access: 11 April 2025). The same approach was taken by Baatsen et al. (2020) using CESM1.0.5 to simulate the Eocene. The background vertical diffusivity  $k_w$ is horizontally homogenous but depends on model depth, and corresponds to  $k_w = vdc1 + vdc2tan^{-1}((|z| - depth)linv),$ where  $vdc1 = 0.524 \text{ cm}^2.\text{s}^{-1}$ ,  $vdc2 = 0.313 \text{ cm}^2.\text{s}^{-1}$ , depth = 1000 m, and linv =  $4.5 \times 10^{-3} \text{ m}^{-1}$ . This equation of background vertical diffusivity follows a Bryan and Lewis (1979) tangential profile, where the upper ocean diffusivity is 0.1 cm<sup>2</sup>.s<sup>-1</sup> and the deep ocean diffusivity is a constant 1 cm<sup>2</sup>.s<sup>-1</sup> below 1000 m. As a comparison, the PI state overall has a lower background diffusivity in the deep ocean, but the bottom-intensified tidal mixing can be up to two orders of magnitude larger than the background mixing. The timestep of the ocean model is reduced from calculating its step every 63 minutes model-time, to 41 minutes model-time. Modifying the timestep of models which use numerical integrations is known to affect their results, as was shown for CAM3 (Mishra and Sahany, 2011; Williamson, 2013), yet it remains generally poorly documented. It is not uncommon for paleoclimate simulations to use shorter computational time steps, as the ocean model in particular can generate numerous numerical instabilities which can lead to an early crash of the model. Our mid-Miocene results are compared to the "best-to-knowledge" PI simulation using modern parameterizations, as this should more accurately resemble the PI climate state. Further discussions regarding the impact of these ocean parameterizations on our PI and Miocene climate will be covered in a future ocean-focused paper.

#### 3 Mid-Miocene control state

## 3.1 Climate

115

Our best-to-knowledge mid-Miocene simulation (Mio\_Ctrl), which includes changes in paleogeography, ice sheet, CO<sub>2</sub> concentration, solar forcing and vegetation, is compared to PI and a similar mid-Miocene simulation using the paleogeography provided for MioMIP2 Phase 1 (Mio\_BurlsGeo), in Fig. 3. Mio\_Ctrl simulates a GMST of 19.8°C, which is 5.4°C warmer than PI (Table 2). This temperature anomaly is in line with other simulations of the mid-Miocene Climate Optimum (MMCO) at lower CO<sub>2</sub> concentrations (Hossain et al., 2023; Wei et al., 2023; Sun et al., 2024), and is around 2°C colder than the mid-Miocene simulations performed with HadCM3L at similar CO<sub>2</sub> concentrations (Burls et al., 2021).

Regionally, our mid-Miocene simulations (Mio\_Ctrl and Mio\_BurlsGeo) both show a substantial warming over the southern high latitudes, linked to polar amplification, and a large cooling pattern over the north Atlantic ocean and northern Europe. The latter is likely connected to an absent Atlantic Meridional Overturning Circulation (AMOC), which is replaced by a Pacific Meridional Overturning Circulation (PMOC). The collapsed AMOC implies a reduced northward heat transport in the North Atlantic and towards the Arctic ocean. The Arctic, in turn, is shallow and has limited exchanges with the Atlantic ocean in these paleogeographies. This likely contributes to the apparent absence of polar amplification in the northern hemisphere temperature signal. A collapse of the AMOC in favor of the PMOC has been seen in other mid-Miocene simulations but remains a rare

135

140

Figure 3. Surface temperatures and total precipitation maps for Mio\_Ctrl, as well as comparisons with PI and Mio\_BurlsGeo

phenomenon (Hutchinson et al., 2025; Naik et al., 2025). This result will be discussed in a connected paper centered on the ocean component of our simulations.

Comparing both paloegeographies, Mio\_BurlsGeo is slightly warmer than Mio\_Ctrl by 0.2°C (Table 2). This difference is of similar amplitude as the difference between the simulation with pre-industrial solar forcing and Langhian solar forcing, which is discussed later on in Section 6.3. Mio\_BurlsGeo was initialized with PI solar forcing as to provide a better comparison with previously published mid-Miocene runs, and this can contribute to the differences in temperatures between both Mio\_BurlsGeo and Mio\_Ctrl. Indeed, a simulation with PI solar forcing (Mio\_noIS\_solPI) shows an additional warming of around 1°C in high latitudes, as later discussed in this study (Section 6.3).

Locally, the paleogeographies differ in bathymetry (Fig. 1), with Getech notably having a shallower Arctic basin, and orography, with mountainous regions and certain plateau regions generally higher in Getech. Due to lapse-rate, differences in surface temperatures between both simulations can simply be a consequence of differences in topography, notably as observed in the

**Table 2.** Summary of global mean surface temperature (GMST) and the anomaly relative to the PI simulation, and percent change of total precipitation ( $\%\Delta p$ ) relative to PI in our simulations.

| Run             | GMST (°C) | GMST anomaly (°C) | $\%\Delta p$ |
|-----------------|-----------|-------------------|--------------|
| PI              | 14.4      | -                 | -            |
| PI_DblCO2       | 17.2      | 2.8               | -            |
| Mio_Ctrl        | 19.8      | 5.4               | 12.9         |
| Mio_2x          | 18.0      | 3.6               | 9.4          |
| Mio_4x          | 21.2      | 6.8               | 15.3         |
| Mio_Ctrl_DblCO2 | 22.7      | 8.3               | -            |
| Mio_2x_DblCO2   | 20.6      | 6.2               | -            |
| Mio_noIS        | 18.5      | 4.1               | 9.0          |
| Mio_noIS_solPI  | 18.7      | 4.3               | 9.5          |
| Mio_BurlsGeo    | 20.0      | 5.6               | 14.0         |
| Mio_noIS_CAM5   | 21.7      | 7.3               | 20.0         |

vicinity of the Rocky Mountains, the Andes, the Tibetan plateau or the Antarctic ice sheet. Getech also includes large changes in the representation of South American Pebas mega-wetlands, which are absent from Burls et al. (2021). Finally, Getech does not include a Greenland ice sheet, but Greenland is instead a slightly elevated plateau.

Similarly as for temperature, we analyse total precipitation in Mio\_Ctrl with respect to PI and Mio\_BurlsGeo. Globally, Mio\_Ctrl is 12.9% wetter than PI (Table 2) and is wetter than mid-Miocene simulations using HadCM3L at 850 ppm of CO<sub>2</sub> (around 10%, Acosta et al. (2024)). Mio\_Ctrl is characterized by drier conditions in the eastern Pacific and wetter conditions in the western Pacific, as well as a common double Intertropical Convergence Zone (ITCZ) bias over the basin (Song and Zhang, 2019). This is in good agreement with the multi-model mean of middle to late Miocene simulations at 560 ppm of CO<sub>2</sub> (Acosta et al., 2024). The north Atlantic region is drier than PI, and similar results are obtained from HadCM3L and CCSM4 simulations (Burls et al., 2021; Acosta et al., 2024).

Compared to Mio\_Ctrl, Mio\_BurlsGeo is slightly wetter, at 14% relative to PI (Table 2). Precipitation patterns are partly affected by the differences in coastlines and topography as seen in Fig. 1. In the tropics, the absence of the Pebas wetlands in Mio\_BurlsGeo results in drier conditions in the northern part of South America. In contrast, the South-East Pacific, North-East Asia and South-East Asia regions are wetter and the latter is, in fact, the largest precipitation change between both geographies. The main characteristic of this region is an enhanced connection between the Pacific and the Indian oceans which is not present in Mio\_Ctrl.

#### 3.2 Vegetation

Biomes based on vegetation distribution are challenging to reconstruct in deep time paleoclimates, notably due to the difficulties in preserving land proxy data and because the plant pollen can be deposited far from their sources. However, vegetation

plays an important role in past climates as it provides feedbacks on moisture and temperature changes, and can be used to reconstruct atmospheric CO<sub>2</sub> concentrations. The mid-Miocene is particularly relevant as it displays the largest flaura change of the Cenozoic and the emergence of modern biomes (Steinthorsdottir et al., 2021).

Coupled climate-vegetation models have been used in past climate simulations (e.g., PlioMIP etc), with relative good agreement with proxy data, albeit large local differences can be seen. Here, we show the simulated vegetation for Mio\_Ctrl (Fig. 4) and compare it to the reconstructed biomes for the mid-Miocene of Steinthorsdottir et al. (2021).

The proxy record indicates a large spread of temperate forests and savanna in mid-latitudes during the mid-Miocene, notably in Europe, north-east Asia and Australia. Although our vegetation model also simulates temperate biomes for Europe and Australia, predominantly composed of temperate shrubs or evergreen temperate forests, we observe Arctic (dominated by Arctic grass) and boreal (dominated by boreal evergreen trees) climates in North East Asia and northern Europe, and almost no vegetation growth above 60°N. On the contrary, the proxy record still indicates (cool) temperate forests, even at latitudes higher than 60°N.

Overall, our predicted vegetation mirrors our model-data temperature map: there is a good agreement in tropical regions between model and proxy data, but our simulated biomes are contracted towards the Equator, with warmer type of vegetation constrained in the lower latitudes and therefore fail to match mid to high latitude biomes as seen in the proxy record. This is likely both a result of and feedback on temperature, as vegetation usually has a lower albedo than bare soil.

There have been different plausible concentrations for atmospheric CO<sub>2</sub> for the mid-Miocene due to differences in methodol-

## 4 Impact of CO<sub>2</sub>

180

185

ogy and calibrations (Burls et al., 2021). The concentration is usually thought to lie between 2 and 3 times PI concentrations, 3 being the concentration discussed for the upcoming MioMIP experimental design, in line with recent high estimates during the MMCO (Rae et al., 2021). Here, we compare Mio\_Ctrl, which has a CO<sub>2</sub> concentration of 3 times PI, to two simulations using 2 times (Mio\_2x) and 4 times (Mio\_4x) PI in Fig. 5, relative to the proxy record. Although there is no evidence that CO<sub>2</sub> concentrations were as high as 4 times PI during the mid-Miocene, these levels are useful for estimating ECS and feedbacks (Farnsworth et al., 2019), which are both important elements for quantifying global and regional responses to CO<sub>2</sub>. In Mio\_2x, the GMST is 18.0°C, which is 3.6°C warmer than PI (Table 2), and even lower than proxy estimates of the colder Late Miocene (Burls et al., 2021). This temperature anomaly is closer to the high end of the simulations of the mid-Pliocene Warm Period (Haywood et al., 2020), where CO<sub>2</sub> is the dominant forcing (Burton et al., 2023). Although there are substantial differences, the mid-Miocene and mid-Pliocene share some similarities in continental configurations, which may also suggest a dominant role of CO<sub>2</sub> as a forcing in the mid-Miocene climate. In Mio\_4x, the GMST is 21.2°C, and its anomaly relative to PI is 6.8°C (Table 2). This is in agreement with proxy estimates for the GMST of the mid-Miocene. Similarly, Acosta et al. (2024) showed that CO<sub>2</sub> concentrations between three and four times pre-industrial concentrations provide the most reduced bias in total precipitation at a global scale, but some regions critically lack proxy sites, and there are few simulations at such high CO<sub>2</sub> concentrations for comparison (Burls et al., 2021).

**Figure 4.** A) Simulated vegetation types in Mio\_Ctrl compared to B) reconstructed biomes for the mid-Miocene, adapted from Steinthors-dottir et al. (2021). In A), the soil is bare if the sum of the contribution of all vegetated areas is less than 10%.

At all CO<sub>2</sub> levels, the mid-Miocene simulations are cold and underestimate surface temperatures reconstructed from proxy data in most mid to high latitudes locations (Fig. 5). The cold pattern simulated in the north Atlantic contributes even more to the temperature bias relative to proxy data. This pattern is connected to a collapse of the AMOC in our model, which is absent at all three CO2 levels. Problematically, while a weak Mid-Miocene AMOC is a persistent feature in climate models (Naik

et al., 2025), the associated colder North Atlantic surface temperature are not supported by proxies (e.g. Sepulchre et al., 2014). Further discussion on the ocean circulation in our simulations is planned for a follow-up paper.

In Mio\_2x, the temperatures are too cold even in tropical locations compared to proxy data (Fig. 5). On the contrary, in Mio\_4x, some of the mid and high latitudes negative biases are dampened, but tropical locations are too warm compared to proxy. These issues affecting polar and tropical warmth are commonly found in other Miocene simulations with various models (Burls et al., 2021), and most mid-Pliocene simulations during PlioMIP1 (Haywood et al., 2011), although there was no systematic bias in PlioMIP2 models (Haywood et al., 2020). Climate models are not tuned for paleoclimates, which contributes to the challenges they face in representing polar amplified temperatures in warm paleoclimates (Burls and Sagoo, 2022). It is therefore not surprising that CESM1.2 shows overly cold temperatures in polar regions, but also a flat warm increase in the tropics in response to CO<sub>2</sub> changes.

Figure 5. Surface temperatures at 2 (Mio\_2x), 3 (Mio\_Ctrl) and 4 (Mio\_4x) times pre-industrial  $CO_2$  concentrations relative to the proxy record, at proxy sites and zonally. Proxy sites are rotated on the mid-Miocene geography with the tectonic model of Getech Plc.

# 5 Equilibrium Climate Sensitivity

ECS is widely regarded as one of the most critical metrics for characterizing future climate change (Forster et al., 2021; Huusko et al., 2021). However, due to the difficulty of constraining it, several independent methods have been used, notably including evidence from the paleoclimate record (e.g. Crucifix, 2006; Hargreaves et al., 2012; Rohling et al., 2012; Renoult et al., 2020; Sherwood et al., 2020). Some of these methods involve directly calculating the radiative forcing and global temperature of a past climate from the proxy record (e.g Rohling et al., 2012; Sherwood et al., 2020; Tierney et al., 2020), or emergent-constraint approaches, which are statistical methods connecting proxy reconstruction with the outputs from several models, usually under the umbrella of a MIP, such as for the LGM under PMIP (Hargreaves et al., 2012; Renoult et al., 2020, 2023) or PlioMIP (Hargreaves and Annan, 2016; Haywood et al., 2020; Renoult et al., 2023).

The ECS of a paleoclimate and how it relates to modern ECS is not necessarily obvious. The effects of "fast" climate feedbacks (i.e. clouds, surface albedo, water vapor, lapse-rate and Planck feedbacks) need to be disentangled from "slow" climate feedbacks (notably carbon cycle, ice sheets and aerosols feedbacks), where the latter affects temperature on geological time-scale and usually connect to a different concept, the Earth system sensitivity (Rohling et al., 2012). Climate feedbacks also depend on the background climate and the forcing imposed, which can lead to high ECS in warm, high-CO<sub>2</sub> paleoclimates (Caballero and Huber, 2013; Anagnostou et al., 2020). Those aspects result in important challenges when using evidence from new paleo-MIP to deduce ECS (Rohling et al., 2012; Renoult et al., 2023).

A way to approach these issues is to estimate ECS directly from a paleoclimate simulation, and compare it to the ECS in a modern simulation in the same model(e.g. Farnsworth et al., 2019). Here, we have performed two simulations: Mio\_Ctrl\_DblCO2, which corresponds to a doubling of CO<sub>2</sub> from Mio\_Ctrl, and Mio\_2x\_DblCO2, which is a doubling of CO<sub>2</sub> from Mio\_2x. Further details on the methodology are provided in Section 2. We report those simulations, as well as comparison to a doubling of CO<sub>2</sub> concentration from PI climate in Fig. 6.

In Mio\_Ctrl\_DblCO2, ECS is 2.9°C (2.5 – 3.3°C, 95% predicted interval, Fig. 6-B), in comparison to 2.8°C from PI conditions with this model (Fig. 6-A). From a lower CO<sub>2</sub> concentration, Mio\_2x\_DblCO2, ECS is 2.6°C, which, similarly as the estimate from PI climate, is also within the 95% predicted interval for the ECS of Mio\_Ctrl\_DblCO2. Even when considering the higher background CO<sub>2</sub> concentration, and consequently higher background mean temperature of the mid-Miocene, these estimates tend to indicate that the PI-based ECS of CESM1.2 can characterize the Miocene climate, or that ECS estimates obtained from the Miocene climate using CESM1.2, for instance in an emergent constraint approach, could be representative of its modern-based ECS. Previous studies on the Miocene have also estimated its ECS to be close to 3°C, and in general relatively close to the ECS estimated for modern climate (Tong et al., 2009; Rohling et al., 2012; Bradshaw et al., 2015; Brown et al., 2022), in comparison to some older warm paleoclimates (e.g. the Eocene) where ECS can reach high values (Anagnostou et al., 2020).

**Figure 6.** Top-of-atmosphere energy flux (W.m<sup>-2</sup> versus global mean surface temperature where A) the simulations temperature are reported relative to the PI temperature and B) the simulation temperatures are reported relative to their initial state (PI or mid-Miocene). The stars are estimates of the long-term near-equilibrium temperatures using ordinary least squares linear regression.

### 6 Additional sensitivity studies

Here, we investigate the impact of the Antarctic ice sheet, atmospheric model, and solar constant on the mid-Miocene temperature and precipitation. The sensitivity runs for the atmospheric model and the solar constant were performed without the Antarctic ice sheet, thus in this section we use Mio\_noIS, a mid-Miocene simulation without Antarctic ice sheet as the control simulation around which the sensitivities are tested. Global temperature anomalies and percent change of total precipitation relative to PI are reported in Table 2.

## 245 6.1 Impact of atmospheric model

250

Model parametrizations play an important role on climate. In the Pliocene Modelling Intercomparison Project Phase 2 (PlioMIP2), three versions of CCSM4 simulated the mid-Pliocene. The global mean surface temperature difference between the warmest CCSM4 and the coldest CCSM4 was as large as the global mean surface temperature difference between the coldest CCSM4 and pre-industrial climate (Haywood et al., 2020). CCSM4 is a model of the CESM model family (i.e. built on earlier versions of submodel components), such as CESM1.2 which we are using for our Langhian simulations. Throughout the model family

**Figure 7.** Surface temperature maps for the different sensitivity experiments, using the no-ice-sheet (Mio\_noIS) experiment as a control state. Note the differences in scale.

Figure 8. Total precipitation maps for the different sensitivity experiments, using the no-ice-sheet (Mio\_noIS) experiment as a control state

history, there have been substantial changes in parametrizations, with the most recent CESM2 having one of the largest ECS of modern climate model ensembles (Gettelman et al., 2019).

Compared to CAM4 with an ECS of 3.2°C at PI (Gettelman et al. (2012), which differs from the estimated 2.8°C of this study in methodology), CAM5 has an ECS of 4°C, mostly due to more positive cloud feedbacks, in particular in the mid latitudes (Gettelman et al., 2012). Logically, this leads to the Langhian state using CAM5 being warmer than when using CAM4, with a global mean surface temperature of 21.5°C and a temperature difference to PI of 7.3°C. Most of the extra warming in CAM5 relative to CAM4 comes from high latitude regions, particularly in the northern high latitudes (Fig. 7). Compared to PI, the simulation using CAM5 is colder in the North Atlantic (Appendix B), and the mid-latitudes and the North Pacific are slightly colder than the mid-Miocene simulation using CAM4. The former is a consequence of the AMOC shutting off, similarly as in the simulations using CAM4, whereas the latter seems connected to the PMOC shutting off, influenced by enhanced precipitation when using CAM5. Although the differences in geography between PI and the Miocene cannot be ignored, the warmer high latitudes are in line with the changes in mid to high latitudes clouds of CAM5 compared to CAM4, where the overall cloud radiative forcing is substantially less negative, which would lead to a warming of those regions. Compared to previous studies, the simulation using CAM5 is almost as warm as HadCM3L using a previous iteration of the Getech paleogeography (Burls et al., 2021).

Similarly as for temperature, changing the atmospheric model from CAM4 to CAM5 has a considerable effect on precipitation. Globally, we observe an increase to total precipitation of 20.0% relative to PI, which is also 11% larger than our CAM4 Langhian simulation without ice sheet (Table 2). Overall, there is a tight coupling between the global mean temperatures and global precipitation changes across our different simulations, as shown in Fig. 9. This is to be expected from Clausius-Clapeyron relation, and it is an indicator of the dominant role of global temperature change on global precipitation, whereas the different sensitivities tested in this study (e.g. topography, atmospheric model), are more likely to affect regional precipitation. For instance, as earlier mentioned, CAM5 saw substantial changes in mid to high latitudes cloud parametrizations and microphysics scheme relative to CAM4, where we observe much wetter conditions in those regions with CAM5 in our simulations (Fig. 8), particularly in the Southern hemisphere and the northern Pacific ocean. This is at first not expected because CAM5 saw reduced non-convective precipitation relative to CAM4 (Chen and Dai, 2019), so the increase in high latitude precipitation is more likely a response of CAM5 to CO<sub>2</sub> rather than differences in parametrizations between both atmospheric models. On the contrary, CAM5 displays enhanced convective precipitation relative to CAM4 (Chen and Dai, 2019), and over the equatorial Pacific ocean, a wetter double ITCZ band is observed in future scenarios using CAM5 compared to CAM4 (Meehls et al. 2013), which could contribute to the substantial wettening of the equatorial Pacific ocean in Mio noIS CAM5. Drier conditions are observed over North America and in the eastern subtropical Pacific ocean. Contrarily to simulations using CAM4, the Pebas wetlands has a less pronounced impact on South American precipitation.

### **6.2** Impact of ice sheet

Geological evidences show a substantial retreat of the Antarctic ice sheet during the early to mid-Miocene, in particular during the MMCO (Gasson et al., 2016). In the two paleogeographies used in this study, we report a reduction in Southern hemi-

**Figure 9.** Global temperature change (°C) versus global precipitation change (%) relative to PI across all our mid-Miocene simulations (dots) initialized from default model conditions.

sphere land ice area (which also include Andean glaciers) of 18% using the paleogeography of Getech, and 31% using the paleogeography of (Burls et al., 2021) relative to our PI state. For the Northern hemisphere, there is no land ice in the Getech paleogeography, and a reduction of 80% using the paleogeography of (Burls et al., 2021). The extent of the land ice is shown in Fig. 2.

Through modification of the surface albedo, the topography and the atmospheric and ocean circulations, ice sheets have a substantial impact on the climate (Gasson et al., 2016). Here, we test the impact of the Antarctic ice sheet on the mid-Miocene climate by comparing to Mio\_Ctrl a simulation where we removed the ice sheet and jointly flatten the Antarctic continent (Mio\_noIS). Although there is a net reduction in the land ice area in our mid-Miocene simulations compared to PI, we would still expect the presence of ice sheets in Mio\_Ctrl to cool down the climate relatively to Mio\_noIS, notably through the increase of surface albedo. Instead, implementing the Antarctic ice sheet in Mio\_Ctrl warms up the global climate by 1.4°C and makes it 3.9% wetter compared to Mio\_noIS (Table 2).

305

320

When comparing both cases with and without ice sheet in Fig. 7, we observe that the presence of the ice sheet locally cools down Antarctica by more than 20 degrees at its highest point. However, there is also a substantial warming in the Arctic ocean and over the northern hemisphere continental masses, as well as a smaller warming over the global ocean. Mio\_noIS shows almost no warming at all in the Arctic region relative to PI, which differs from Mio\_Ctrl. Over the oceans, mid and low latitude regions are drier or wetter in Mio\_Ctrl if they are respectively dry and wet in Mio\_noIS. High latitudes tend to be wetter in Mio\_Ctrl, except directly over the Antarctic ice sheet. A substantial wettening of Eastern Africa can be observed in Mio\_Ctrl, as well as Southern Asia.

Atmospheric and oceanic heat transport are similar between the simulations, indicating that the higher Northern Hemisphere temperatures are not due to enhanced heat transport from the south (Appendix A). Instead, the radiation budget in the northern high latitudes suggests that the warming could be a result of a complex interplay between cloud and ice fraction forces and feedbacks. A detailed investigation of these mechanisms is beyond the scope of this study, but we briefly discussed it in Appendix A, as it could highlight some processes which affect polar amplification in warm paleo-simulations.

### **6.3** Impact of solar constant

Of all the forcing applied to our mid-Miocene simulations, the change from PI solar constant to mid-Miocene solar constant has the least impact. Mio\_noIS\_solPI is around 0.2°C warmer than Mio\_noIS (Table 2), whereas most of the warming comes from high latitudes where the temperatures are 1°C warmer (Fig. 7). The small amplitude of this change is expected, as the mid-Miocene solar constant is estimated to be 0.1% less than PI solar constant (Gough, 1981). As a comparison, a doubling of atmospheric CO<sub>2</sub> (around 3.7 W.m<sup>-2</sup>, Myhre et al. (1998)) is roughly equivalent in forcing to an increase of 2.25% of solar constant, assuming a planetary albedo of 0.7. Therefore, the change of solar constant is mostly negligible, but could be partially responsible for the high latitudes of Mio\_BurlsGeo being warmer than Mio\_Ctrl which uses a mid-Miocene solar constant.

## 7 Conclusions

In this study, we have developed a set of simulations using a new and unpublished geography for the Langhian, using CESM1.2. Research centered on the mid-Miocene has shown a potential for it to be a near-future analog, yet uncertainties remain on atmospheric CO<sub>2</sub> concentration and non-CO<sub>2</sub> forcing, and consequently the scale of the warming. In particular, models fail at simulating high latitude warming that is observed in the proxy record. The mid-Miocene experiments performed in this study aimed to bring a new light onto those questions.

Variations in CO<sub>2</sub> concentrations for the mid-Miocene have a substantial influence on its estimated temperatures. Between the high end (4 times PI concentration) and the low end (2 times PI concentration) of tested concentrations, we find a difference of 3.2°C on mid-Miocene global mean surface temperature, suggesting a dominant role of CO<sub>2</sub> forcing on the mid-Miocene climate, as also previously observed in the case of the Pliocene (Burton et al., 2023). The mid-Miocene is also sensitive to non-CO<sub>2</sub> forcing, such as the land-sea mask, ice sheet and solar forcing, but the variations tested for those components had a lower influence on the mid-Miocene temperature than CO<sub>2</sub>. Finally, we identified that switching the atmospheric model

from CAM4 to CAM5, which has a substantial influence on cloud feedbacks, leads to a warmer mid-Miocene by up to 3.2°C, which highlights the critical need of improving model climate feedbacks, as much as it is to better constrain climate forcing in paleoclimates.

Similarly as for the LGM (Hargreaves et al., 2012; Renoult et al., 2020) and the Pliocene (Hargreaves and Annan, 2016; Haywood et al., 2020; Renoult et al., 2020), the status of the mid-Miocene as a potential analog for future climate can be used to quantify ECS. In this study, we estimated that the ECS of CESM1.2 at the mid-Miocene is 2.9°C (2.5 – 3.3°C, 95% prediction intervals), which is close to the modern-based estimate of ECS of CESM1.2 (2.8°C). From this short glance on ECS in CESM1.2, as well as previous work on Miocene ECS (Tong et al., 2009; Rohling et al., 2012; Bradshaw et al., 2015; Brown et al., 2022), we believe there is motivation in supporting ECS estimates using Miocene evidence, particularly as Miocene modelling is moving towards an official PMIP design. Although some issues might arise from uncertainties in the proxy record or model disagreements, the Pliocene, which is the following warm epoch on the geological time scale, has shown that recent warm paleoclimates, such as here for the mid-Miocene, can provide robust constraint on ECS despite those issues (Hargreaves and Annan, 2016; Haywood et al., 2020; Renoult et al., 2023). Additionally, emergent constraint framework can be used to identify disagreements between models, and between model and data (Renoult et al., 2023). Thus, estimates from the Miocene could either support existing methods using paleoclimate evidence (Sherwood et al., 2020; Cooper et al., 2024), or be used within a new emergent constraint framework as previously done for the LGM (Hargreaves et al., 2012; Renoult et al., 2020) or the Pliocene (Hargreaves and Annan, 2016; Haywood et al., 2020; Renoult et al., 2023).

Compared to the proxy record, mid-Miocene simulations are usually either too cold globally, too cold at high latitudes, or both (Burls et al., 2021). Over different sensitivity experiments, we find the same issues using CESM1.2, which arise from a weak polar amplification. Simulated high latitudes temperatures underestimate the proxy record by as much as 20°C, and warm vegetation biomes are more contracted towards low latitudes. However, high CO<sub>2</sub> levels and better representation of high latitude cloud parametrizations produce a better match with high latitude proxy sites, albeit they also overestimate low latitude temperatures. Higher CO<sub>2</sub> concentrations (between three and four times pre-industrial) also provided a better match with precipitation proxy reconstruction, which highlights the importance of high CO<sub>2</sub> concentrations for representing the mid-Miocene climate (Acosta et al., 2024). A mid-Miocene state close to reality is likely to include a combination of different sensitivities tested in this paper. Future research should have a focus on high latitude climate feedbacks as to enhance polar amplification factor in models, which has been a recurring problem in paleoclimate simulations and could affect future simulations. Finally, it is likely that climate models will require higher CO<sub>2</sub> concentrations (three to four times PI levels) than previous estimates for the mid-Miocene (two to three times PI levels, (Rae et al., 2021)) as to compensate for the low polar amplification aforementioned, but also the globally too cold Miocene climate. Refining these aspects of the mid-Miocene would bridge the gap between proxy reconstructions and climate models, which in turn could improve our understanding of climate sensitivity, feedbacks and future climate change.

Data availability. The temperature and precipitation outputs of the simulations of Table 2 and the vegetation data of Mio\_Ctrl are available at https://doi.org/10.5281/zenodo.17303799

Author contributions. The idea of the study was conceived by MR and AdB. MR performed all simulations, analyses and figures. The paper was written by MR, AdB, and EB.

Competing interests. The authors declare that they have no conflict of interest.

Acknowledgements. The computations and data handling were enabled by resources provided by the National Academic Infrastructure for Supercomputing in Sweden (NAISS) partially funded by the Swedish Research Council through grant agreement no. 2022-06725. Getech Group Plc. provided the Langhian paleogeography and support for running the simulations. AdB received funding from the Swedish Research Council grant 2020-04791.

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

## Appendix A: Impact of ice sheet on heat transport and radiation

When implementing an Antarctic ice sheet in our mid-Miocene simulations, we observe a global warming and a particularly pronounced warming over the Arctic region. The climate feedbacks coupling the Antarctic ice sheet to Arctic warming is not straightforward, and beyond the scope of our study. However, we would like to highlight here a few elements to motivate future research on the topic.

Figure A1. Atmospheric and oceanic heat transports (PW) per latitude between both Mio\_Ctrl and Mio\_noIS

A first hypothesis was that the ice sheet was somehow causing enhanced heat transport to the Arcitc. However, the addition of the ice sheet did not lead to an increased in either oceanic or atmospheric heat transport. (Fig. A1). Instead, the difference between atmospheric, oceanic and total heat transport between the simulations is so small, in particular in northern high latitudes, that it suggests local processes are affecting Arctic temperatures.

We compare three processes which are related to climate feedbacks in the Arctic region between the simulations in Fig. A2: effective albedo, ice fraction and longwave cloud forcing. When implementing the Antarctic ice sheet in Mio\_Ctrl, we observe an overall reduction of surface effective albedo in the Arctic region (over the ocean and land), a reduction in annual Arctic sea-ice and an increase in longwave cloud forcing. All of these elements would favor a warming of the Arctic region compared to Mio\_noIS, by either trapping more heat into the system or decreasing the reflection of shortwaves to space.

From this brief analysis, we are unable to clearly identify the origin of the warming. Climate feedbacks are known to interact with each other, and in particular in the Arctic, there are notable interactions between cloud and sea-ice feedbacks (Kay and

Gettelman, 2009), as well as a large role from temperature feedbacks (Pithan and Mauritsen, 2014), with substantial model differences (Block et al., 2020). Therefore, the problem is left open, and could be relevant for questions of polar amplification in warm paleoclimates.

Appendix B: Additional figures

**Figure A2.** Example of three processes which are tied to climate feedbacks in the Arctic region between Mio\_noIS and Mio\_Ctrl: effective surface albedo, ice fraction and longwave cloud forcing.

Figure B1. Map of surface temperature differences between Mio\_noIS\_CAM5 and PI. Note that the color mapping is non-linear.