# Peer review of "Shaping the mid-Miocene warmth: a sensitivity study on paleogeography, CO2 and model physics"

_EGUsphere, 2025_

## Referee Comment (RC1)

**Review: Shaping the mid-Miocene warmth: A sensitivity study on paleogeography, CO2 and model physics**

Authors: Martin Renoult, Agatha de Boer, and Ellen Berntell

**Abstract**

Line 4 – "...model temperatures are consistently too low" implies that this is *solely* model error, but it is likely a combination of model error and proxy error/bias.

Line 4 – "using a **previously** unpublished geography...", as this will be published work.

Line 7 - I don't fully understand the sentence starting "GMST varies by up to...". Are these control experiments that just alter  $CO_2$ , and are otherwise representative of the pre-industrial?

Line 9 – Could you provide an approximate value or range for local warming?

Lines 11/12 – Could you provide an uncertainty range for the ECS estimate, and also for the "modern-based estimates"?

Line 16 – I would suggest a rephrasing of "...cold modelled Miocene climate" as it is a bit confusing given you are speaking about the *warm*[er than present] Miocene.

**1. Introduction**

Line 18 – The use of "stage" rather than "age" is usually when referencing the rock record only, so I would prefer to see "age" used here.

Line 22 – Similar to above, please use the correct terminology of "age" rather than "epoch" here.

Line 28 – ECS needs to be written out in full on first use, rather than as "equilibrium ECS (ECS)". GMST can also be defined at the end of this line.

Line 32 – The Eocene is an "epoch", not a "period".

Line 33 – Grouping the Pliocene with recent interglacials and the Eocene here misleads the reader, because the Pliocene does have similar CO2 concentration to modern (~400 ppm). This needs to be rephrased. I see value in explicitly stating the Miocene CO2 concentration (with uncertainty range as required) and then relating this to specific instances of near-future estimates (e.g. the estimate for e.g. 2040 under RCP2.6 will be very different to e.g. 2060 under RCP8.5). This is particularly

relevant because present-day estimates of ~420 ppm are significantly closer to Pliocene estimates than the ~850 ppm estimate used for the Miocene here.

Line 36 – I'd consider citing Burls et al. (2021) when referencing the first MioMIP: Simulating Miocene Warmth: Insights From an Opportunistic Multi-Model Ensemble (MioMIP1) - Burls - 2021 - Paleoceanography and Paleoclimatology - Wiley Online Library.

Line 38 (and similarly in Section 4) – In this paragraph on the 'high latitude paradox', I'd add a more explicit note to say that this is likely a combination of both model error and proxy error/bias (rather than just model error). You could also reference Tindall et al. (2022): CP - The warm winter paradox in the Pliocene northern high latitudes.

Line 45 – "...a **previously** unpublished paleogeography".

Line 48 – "...constraining modern-day ECS..."?

Line 49 – I would like to see a little more comment on the value that these extra sensitivity studies bring and/or why they are needed.

**2. Methods**

Line 51 – The phrasing of "Langhian or mid-Miocene" is open for erroneous interpretation. Pick one term to use and be consistent throughout the paper.

Line 52 – Is there a model description paper for CESM1.2 that you can cite here?

Table 1 caption – How long were the experiments run for that are not marked by \*?

Line 55 – LGM needs to be defined here as it's the first use.

Line 60 (and 66, 120) - Should "MioMIP2 Phase 1" read "MioMIP1"?

Line 65 – The sentence starting "A comparison of the..." appears incomplete.

Line 77 – Is there a citation for the tectonic model of Getech Plc?

Figure 2 – For readers not accustomed to viewing Miocene palaeogeography, some additional shading to mark which area is land and which area is ocean could be beneficial.

Line 100 – NCAR needs to be defined on first use.

**3. Mid-Miocene control state**

**3.1. Climate**

Line 130/131 – Unclear what is meant by the PMOC remaining "a rare phenomenon", you've just said that it's been simulated multiple times. Do you mean that there is no/little proxy evidence for a PMOC?

Figure 3 and caption – I would like to see individual labels here (a), b) etc.) so that you can add more detail to the caption and refer to specific plots in the main text. Additionally, I'd like to see lat/lon labels for the axes (as in Figure 2) rather than unitless values.

Line 139 – Be cautious how you refer to the different palaeogeographies. Here you refer to "Getech", but in the previous paragraph it is the experiment name. Be consistent, or for the avoidance of doubt use both (e.g. "the Getech palaeogeography used in Mio Ctrl").

Table 2 – Unless the raw GMST values are used elsewhere in the paper then I don't see the benefit of including both raw GMST and anomaly relative to PI (especially since precipitation is only shown as an anomaly).

Line 152 – Is this 14% wetter than Mio\_Ctrl or PI? It could be read as either from this sentence alone.

**3.2. Vegetation**

Line 162 - Typo on "...largest flora change..."

Line 164 – All models in PlioMIP2 (except COSMOS (Stepanek et al., 2020)) use the static vegetation reconstruction of Salzmann et al. (2008), so this needs to be removed or rephrased to avoid misleading the reader.

Figure 4 – It is quite difficult to quickly compare these figures as the colours and descriptions are not the same between figures. Could the colour scale of A) be adapted to better aid this comparison? Additionally, I would like to see the label for B) reading "mid-Miocene" for consistency with the remainder of the text (to avoid confusion for newcomers to the topic).

**4. Impact of CO2**

Line 179 – Add a value to quantify "PI concentrations". Is this 280 ppm? A more specific CMIP6 value?

Line 181 – Similarly, add value(s) to quantify "high estimates during the MMCO" from Rae et al. (2021).

Lines 185 and 186 – I feel you need to clarify that you are looking at mid-Miocene in Mio 2x. The reference to the colder Late Miocene is currently a bit confusing.

Lines 187-189 – The sentence "Although there are..." feels a bit unsubstantiated. Can you add more comment here? Expand on the "substantial differences", and why the similarity in continental configuration implies a dominant role in CO2. What about the other non-CO2 factors considered in Burton et al. (2023)?

Line 190 – To develop your discussion, I think you can add a more explicit comment that Mio\_4x being in agreement with proxy estimates implies that this is likely a good/representative estimate of CO2 for the mid-Miocene.

Figure 5 – As with Figure 3, I would like to see lat/lon labels for these axes where appropriate and the addition of a), b) etc. so you can refer to specific plots in the main text. Additionally, an axis label (i.e. Temperature) for the bottom right panel, and clarification of the calculation in the caption (e.g. Mio 2x – PI), would be useful.

**5. Equilibrium climate sensitivity**

Line 215 – Although PlioMIP and MioMIP have been defined, it is probably best to define "MIP" here.

Line 220 – Lunt et al. (2010) is arguably the seminal piece on Earth System Sensitivity so should also be cited: Earth system sensitivity inferred from Pliocene modelling and data | Nature Geoscience

Line 227 – I would like to see some extra detail added to "We report those simulations...", or remove this sentence and just refer to it in the next sentence as you already do.

Lines 229 and 230 – For the reader, it would be helpful to clarify what the CO2 concentration is in ppm for each experiment here. It's quite easy to get lost – one can easily assume that "...2x DblCO2" is a higher concentration than "Dbl CO2".

Line 236 – I think it is a missed opportunity not to provide a current estimate of modern ECS, e.g. the range presented in IPCC AR6. This is helpful context for the reader, especially those who are new to this area.

Figure 6 – I spent time looking for a black star as is visible in the legend, it needs to be clearer that there is a star for each of the experiments (one in each colour).

**6. Additional sensitivity studies**

Line 242 – Add a comment on why these sensitivity runs were completed without an Antarctic ice sheet (or add to methods and refer to the relevant section here).

**6.1. Impact of atmospheric model**

Lines 247-249 – This is an important point to make but it would benefit from the additional detail of specific model names. Which CCSM4 version is the warmest? Coldest? For clarity for the reader, I think it would be better to at least refer to "CCSM4 **version**" rather than just e.g. "coldest CCSM4".

Line 250 – Are you trying to say that CESM1.2 is part of the CESM2 model family here, or that it is a version of CCSM4? It is not clear either way.

Lines 267-268 – Adding experiment IDs would be useful here, particularly as you refer to Table 2.

Line 271 – I'm confused that you're referencing the atmospheric model as an example here, in a section about the impact of the atmospheric model. Is the model affecting global precipitation, or regional?

Figures 7 and 8 – As in other figures, please add lat/lon labelling to the axes. I would also like to see the colour bar labels edited to say "surface temperature **anomaly**" or "total precipitation **anomaly**" where appropriate, especially as this is not highlighted in the (relatively brief) caption. Readers should also be encouraged to note the difference in scale in the Figure 8 caption (as they are for Figure 7).

Figure 9 – I think there would be value in labelling which experiment is which dot here, or using different symbols and displaying the experiment IDs in a legend. This would add more depth to the analysis and aid further discussion.

Overall comment – I would like to see a bit more discussion here. What are the implications of the choice of atmospheric model being important (e.g. for MioMIP)?

**6.2. Impact of ice sheet**

Line 297 – By "at its highest point", do you mean highest point in elevation, or greatest temperature anomaly? This needs to be clearer.

**6.3. Impact of solar constant**

Overall comment – Again, a bit more discussion would be valuable. E.g. could this be a lesson going into MioMIP or future Miocene modelling efforts?

**7. Conclusions**

Line 317 – Like elsewhere, I'd phrase as "...new and **previously** unpublished paleogeography".

Line 319 – I think you can go further than just saying "...consequently the scale of the warming" here. This underlies the possibility of using the Miocene as an analogue, and these uncertainties are rightly mentioned alongside a suggestion of analogy. Even if temperature is shown to be similar to future projections, if CO2 is seen to be non-dominant in the Miocene, can we really call it an analogue? What about the AMOC shutdown? There is some discussion around the need for critical thinking in this area in Burton et al. (2023: CP - On the climatic influence of CO2 forcing in the Pliocene), Oldeman et al. (in review: A Framework for Assessing Analogy between Past and Future Climates by Arthur M. Oldeman, Lauren Burton, Michiel L. J. Baatsen, Henk Dijkstra, Anna von der Heydt, Aisling Dolan, Alan M. Haywood, Daniel Hill, Julia Tindall :: SSRN), and Burton et al. (2025: An assessment of the Pliocene as an analogue for our warmer future - ScienceDirect).

Lines 351-352 and 355-357 – Representing mid-Miocene climate... assuming that the high values indicated by proxies are correct. There is likely a middle ground here where both models and proxies are biased. Turning up the CO2 to unevidenced levels in the models is not the right approach!

**Appendix A: Impact of ice sheet on heat transport and radiation**

Reference to Appendix A comes after reference to Appendix B in the main text, so they need to be switched around (i.e. Appendix A becomes Appendix B, and B becomes A).

Line 507 – Typo on "Arctic".

Line 508 - "increased" should read "increase".

**Appendix B: Additional figures**

The appendix only contains one additional figure, so I would like to see the header rephrased and perhaps a sentence or two describing the figure.